# Exploring bacterial key genes and therapeutic agents for breast cancer among the Ghanaian female population: Insights from *In Silico* analyses

Md. Kaderi Kibria[1,2]*, Md. Ahad Ali[1,3], Md. Nurul Haque Mollah[1]*

1 Bioinformatics Laboratory, Department of Statistics, University of Rajshahi, Rajshahi, Bangladesh,
2 Department of Statistics, Hajee Mohammad Danesh Science and Technology University, Dinajpur,
Bangladesh, 3 Department of Chemistry, University of Rajshahi, Rajshahi, Bangladesh

* kibria.stt@tch.hstu.ac.bd (MKK); mollah.stat.bio@ru.ac.bd (MNHM)

## Abstract

Breast cancer (BC) is yet a significant global health challenge across various populations including Ghana, though several studies on host-genome associated with BC have been investigated molecular mechanisms of BC development and progression, and candidate therapeutic agents. However, a little attention has been given on microbial genome in this regard, although alterations in microbiota and epigenetic modifications are recognized as substantial risk factors for BC. This study focused on identifying bacterial key genes (bKGs) associated with BC infections in the Ghanaian population and exploring potential drug molecules by targeting these bKGs through in silico analyses. At first, 16S rRNA bacterial sequence data were downloaded from NCBI database comprising 520 samples from BC patients and 442 from healthy controls. Analysis of 16S rRNA-Seq data showed significant differences in bacterial abundance between BC and healthy groups and identified 26 differential genera with the threshold values at $|\log_2 FC|>2.0$ and $p$-value$\leq 0.05$. It was observed that two genera *Prevotella* and *Anaerovibria* are significantly upregulated in BC patients and others are downregulated. Functional analysis based on all differential genera identified 19 MetaCyc signaling pathways, twelve of which were significantly enriched in BC patients by containing 165 genes Top-ranked 10 genes *mdh*, *pykF*, *gapA*, *zwf*, *pgi*, *tpiA*, *pgk*, *pfkA*, *ppsA*, and *pykA* were identified as BC-causing bacterial key genes (bKGs) through protein-protein interaction network analysis. Subsequently, the bKG-guided top ranked 10 drug molecules Digitoxin, Digoxin, Ledipasvir, Suramin, Ergotamine, Venetoclax, Nilotinib, Conivaptan, Dihydroergotamine, and Elbasvir were identified using molecular docking analysis. The stability of top-ranked three drug-target complexes (Digitoxin-*pykA*, Digoxin-*mdh*, and Ledipasvir-*pgi*) were confirmed through the molecular dynamics simulation studies. Therefore, these findings might be useful resources to the wet-lab researchers for further experimental validation on bacterial therapies against BC.

**Data Availability Statement:** The raw 16S rRNA sequence profile data set analyzed in this study is publicly available. It can be freely downloaded from

the online NCBI database with bio-project number PRJNA658160 (https://www.ncbi.nlm.nih.gov/bioproject/PRJNA658160).

**Funding:** The author(s) received no specific funding for this work.

**Competing interests:** NO authors have competing interests

## 1. Introduction

Breast cancer (BC) poses a significant global health challenge, with varying incidence and mortality rates across different populations [1, 2]. It is the leading causes of cancer-related deaths in women globally [3]. According to the World Health Organization (WHO), approximately 2.3 million women were diagnosed with BC in 2022, leading to 670,000 deaths worldwide [4]. The number of new cases is projected to rise to 3.06 million annually by 2040 [4]. In Africa, BC is the most common cancer among women, representing about 30% of all cancer cases, with an estimated incidence rate of 25% in sub-Saharan Africa [4, 5]. Women of African ancestry, especially those in younger and middle age groups had higher BC rates compared to women of non-African ancestry [6, 7]. This increase rate is likely due to a combination of modifiable and non-modifiable factors, including genetic mutations, DNA methylation, microbial infections and epigenetic modifications, obesity, the adoption of a westernized lifestyle, delayed childbearing, shorter breastfeeding periods, and increased use of oral contraceptives [8–12]. These risk factors can modify the microbial community, potentially linking it to the onset and progression of BC by affecting T cells, neutrophils, and various inflammatory factors [13–19]. Additionally, microbial-genes interact with host-genes for BC development and progression through various mechanisms that involve immune modulation, inflammation, and epigenetic changes [20, 21]. Microbiome release a variety of metabolites that can influence host cellular processes [22]. These metabolites can enter the bloodstream and affect distant tissues, including breast tissue, thereby influencing the expression of host genes [23]. Some microbial metabolites can act as histone deacetylase (HDAC) inhibitors, leading to changes in chromatin structure and influencing the expression of genes involved in cell cycle regulation, apoptosis and DNA repair [24]. These epigenetic modification can promote gene expression, potentially leading to oncogenic transformations in breast tissue and the development of BC [25]. Thus, microbial genome stimulates host-genome for the incidence of BC.

There are several studies on host-genome for exploring drug targets and agents for BC [26–31]. However, a little attention has given on microbial genome in this regard. There are some studies on bacterial genome based on 16S rRNA-Seq profiles that explored differentially abundant bacterial compositions between BC and control samples, bacterial taxa and signalling pathways associated with BC [32–41]. Similarly, a study on the Ghanaian population investigated only bacterial diversity between BC and control samples, but did not explore BC-causing bacterial key genes (bKGs) and therapeutic agents [17]. In order to inhibit bacterial pathogens, it is required to detect their pathogenic genes as drug targets. It might be noted here that either host-proteins [42–45] or pathogenic-proteins [46–52] are used as the receptor proteins for disclosing candidate drug-agents, since host-proteins interact with the pathogenic proteins to develop infections [46–52]. Some studies proposed antibacterial agents in order to inhibit bacterial pathogens in BC [14]. However, due to environmental changes, bacterial pathogens may develop resistance to antibacterial drugs. This type of bacterial pathogens may gradually increase due to the limited antibacterial agents [53, 54]. In this case, alternative potential antimicrobial drugs may be required for effective treatment against BC. This study aimed to disclose such BC-causing bKGs based on 16S rRNA-Seq profiles in order to explore candidate drug molecules as the inhibitors of BC-causing pathogenic bacteria for Ghanaian population, since 16S rRNA-Seq profile analysis is able to disclose non-cultured bacteria and novel pathogens [55, 56]. The overall workflow of this study shown in Fig 1.

## 2. Materials and methods

### 2.1 Data source and description

The 16S rRNA bacterial raw sequence data of human stool samples, along with associated metadata, were considered in this study. A total of 962 16S rRNA sequence reads were

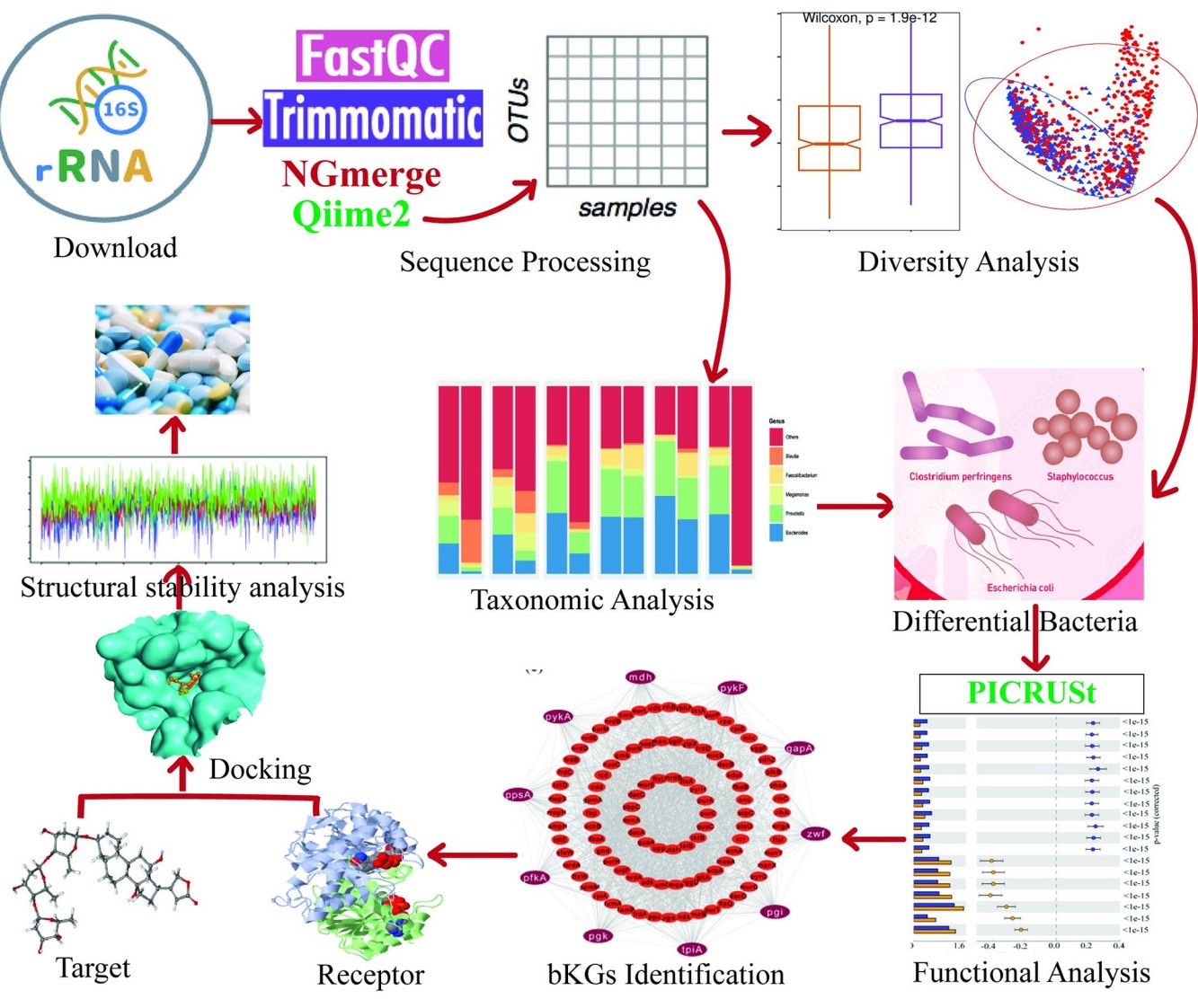

**Fig 1. This is the overall workflow of our study.**

downloaded from the NCBI database, corresponding to the bio-project accession ID PRJNA658160. Among these, 520 samples were from BC patients (case group) and 442 samples were from healthy individuals (control group). All the samples are representative of the population in Ghana. A previous study on the Ghana population showed the association of fecal bacteria with BC and non-malignant breast disease by comparing cancer cases, non-malignant cases, and healthy controls [17]. Using alpha and beta diversity analysis, the study found that fecal bacterial characteristics were strongly and similarly associated to both BC and non-malignant beast disease.

In this study, a total of 198 BC-preventing drug molecules were collected by reviewing 17 published articles (see S5 Table) to repurpose potential drugs against the identified bKGs. To assess the accuracy of the docking performance, 528 hub proteins associated with BC were collected from 100 published articles (**see** S4 Table). Among these, 10 proteins *(BUB1, TOP2A, CDK1, AURKA, CDC20, EGFR, CCNB1, CCNA2, BUB1B,* and *FN1)* were commonly reported in at least 7 of the reviewed articles. These 10 hub proteins were used for docking analysis

against the 198 drugs to validate the results of our proposed protein-docking results. The protein structures were downloaded from the Protein Data Bank (PDB) with the IDs *4a1g*, *1zxn*, *4yc3*, *1mq4*, *4gga*, *2itv*, *1e9h*, *3si5* and *1fnf*, while the drug molecules were downloaded from the PubChem database.

## 2.2 Preprocessing of 16S rRNA-Seq profiles

After downloading the 16S rRNA sequences of BC and control samples from the NCBI database, the data were prepared for comprehensive analysis. Initially, the quality of the raw sequence reads was assessed using FASTQC software to identify issues such as low-quality bases, adapter contamination and overrepresented sequences. Poor-quality reads and adapter sequences were then trimmed using Trimmomatic-0.39 software with default parameters [57], ensuring that only high quality reads were retained for further processing. Following this, the paired-end sequences were merged using Ngmerge (v0.2) software, ensuring a minimum overlap of 5 base pairs and allowed no more than 10% mismatches [58]. The merged sequences were subsequently imported into Qiime2 (version: 2023.9) software [59], a widely used platform for microbiome analysis. Within Qiime2, the sequences were dereplicated using VSEARCH [60], where redundant sequences were collapsed into unique sequence variant. An open-reference clustering approach was then employed, grouping the sequences into operational taxonomic units (OUT) at 97% similarity against the Greengenes database. The resulting OTU table, containing the abundance data for each taxon across samples, was generated and served as the foundation for downstream diversity analyses, including alpha and beta diversity measures, which provide insights into microbial composition and variation between the BC and control groups.

## 2.3 Bacterial diversity analysis

Bacterial diversity analysis was performed by using Qiime2 (Version: 2023.9) and R software (Version: 4.2.2). Within sample diversity (alpha diversity) was evaluated using Chao1, Observed, ACE indices and Good's estimator of coverage [61], calculated through the "phyloseq" R package. Visualization of results employed the ggplot2 R [52, 62]. Statistical significance in alpha diversity between groups was determined using the Wilcoxon rank-sum test [63]. On the other hand, between sample diversity (beta diversity) was assessed using Bray-Curtis distances [64], calculated using "microbiotaProcess" R package. Principle Coordinate Analysis (PCoA) plots were generated to visualize the differences in microbial community compositions across samples [65]. This approach helps reveal patterns, clusters, and the overall variability between BC and healthy groups. Clustering patterns in beta diversity were evaluated using permutation multivariate analysis of variance (PERMANOVA) with 999 permutations, conducted with the "vegan" R package [66].

## 2.4 Taxonomy analysis and identification of differentially abundant bacteria

A Naive Bayesian classification algorithm of the RDP (Ribosomal Database Project) classifier (v2.2) was employed to assign representative sequences to bacterial taxa. Relative abundances of these taxa at the phylum, class, order, family, genus, and species levels were calculated using Qiime2 (version 2023.9). To identify differentially abundant bacterial taxa (DABT), a Zero-Inflated Gaussian mixture model (ZIGMM) was applied to mean group abundance data at a specified *p*-value threshold $p \leq 0.05$ for statistical significance [67]. The Likelihood Ratio Test (LRT) was used to determine the differences in bacterial taxa abundance between groups.

## 2.5 Pathway-based gene identification from DABT

Pathway based BC-causing genes were identified from DABT by using PICRUSt2 (Phylogenetic Investigation of Communities by Reconstruction of Unobserved States). Differentially abundant metabolic pathways (DAMPs) between BC and control groups were identified using STAMP software, which applied Welch's $t$-test to assess their statistical significance. Genes associated with these DAMPs were considered as BC-causing bacterial genes. Protein-protein interaction networks among the identified bacterial genes were analysed using the STRING database to identify BC-causing bacterial key genes (bKGs) and visualized with Cytoscape.

## 2.6 Bacterial key genes (bKGs)-guided drug repurposing

Drug repurposing reduces both time and cost compare to the de-novo approach [68–71]. To identify effective repurposable drugs for inhibiting bKGs, docking interactions between ligands and bKGs-mediated receptors by using AutoDock Vina [72]. Protein structures were sourcd from the Protein Data Bank (PDB), and pre-processed to remove heteroatoms, water molecules, and unused ligands using BIOVIA Discovery Studio [73] and PyMOL [74]. Protein energy minimization was performed with Swiss-Pdb Viewer [75], followed by conversion to PDBQT format and adjustment of grid box size and center using AutoDock tools. Ligands were minimized using Avogadro software with the MMFF94 force field [76], employing conjugate gradient algorithms with specific parameters including a total of 200 steps, state updates every 1 step and energy difference threshold of 0.1 [77]. After minimization, ligands were converted to PDBQT format and the pKa values of ionizable groups in the protein-ligand complexes were estimated at physiological pH 7.0 using PROPKA. Docking analysis was performed with AutoDock Vina (version 1.1.2), using an exhaustive search parameter set to 10. Non-bond interactions were analyzed using BIOVIA Discovery Studio (version 3.0) and PyMOL software (version 2.3).

Molecular dynamics simulations of selected protein-ligand complexes were performed using YASARA software [78] with the AMBER14 force field [79], at 298K, pH 7.4 and 0.9% NaCl. Simulation included neutralization and energy minimization steps, and each simulation ran for 100 ns. Snapshots were analyzed for binding free energy using MM-Poisson–Boltzmann Surface Area (MM-PBSA) methods [80, 81], where binding energy was calculated as follows.

$$\text{Binding Energy} = E_{potRecept} + E_{solvRecept} + E_{potLigand} + E_{solvLigand} - E_{potComplex} - E_{solvComplex}$$

The YASARA macro was used to calculate MM-PBSA binding energies, where positive values indicate stronger binding [82]. Additionally, simulation snapshots were used to compute root-mean-square deviation (RMSD), radius of gyration, and solvent-accessible surface area (SASA) [83–87].

## 3. Results

### 3.1 Preprocessing of bacterial 16S rRNA-Seq profiles data

In this study, a total of 962 individual samples from Ghana were analyzed, comprising 520 samples from BC patients and 442 samples from healthy individuals, as described in the methods section. Stool samples were collected, and the V4 region of the 16S rRNA sequence was sequenced to profiles bacterial communities, with the sequences deposited in the NCBI database. Trimmomatic software was used to remove low-quality reads, retaining approximately 89% of the sequences. After dereplication, 2,149,741 features were identified across all samples. Open reference clustering at 97% similarity against the Greengenes database resulted 98,108

OTUs. The OTU table was filtered with a minimum feature count of 5 and a frequency threshold of 1000, resulting in 98,086 OTUs across 910 samples, with an average of 18,268 OTUs per sample. Subsequently, the OTU table was rarefied to the lowest total number of observed individuals across all samples, and these rarefied OTUs were used for diversity analysis in downstream investigations.

## 3.2 Bacterial diversity analysis

Alpha and beta diversity were used to evaluate differences in bacterial composition between BC patients and healthy controls. Alpha diversity was measured using the Chao1, Observed, and ACE indices, revealed a significant decrease in bacterial diversity in BC patients compared to healthy individuals (see **Fig 2(A)–2(C)**, $p$-value<0.01). Good's coverage estimates ranged from 0.864 to 0.996 in BC patients and from 0.821 to 0.9916 in healthy individuals ($p$-

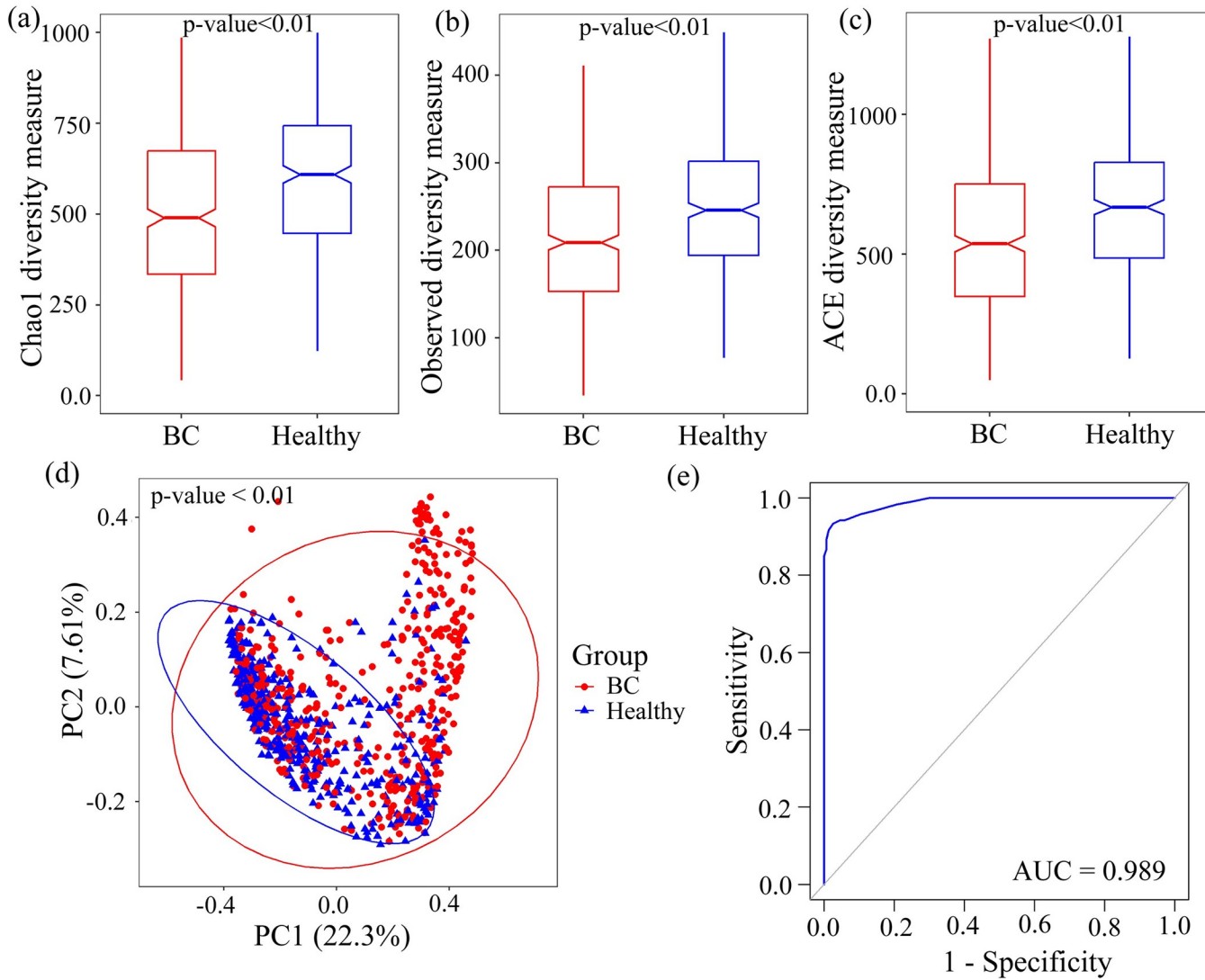

**Fig 2. Discrepancy in bacterial profiles between BC patients and healthy controls.** (a,b,c) Chao1, Observed and ACE indices of diversity in BC patients and healthy samples. (d) PCoA plot of Bray-Curtis distances, with sample groups indicated by two different colors and percentage of diversity captured by two coordinates. (e) ROC analysis results evaluating the predictive value of OTUs in distinguishing BC and healthy groups.

value = 1.5e-10), indicated a comprehensive estimation of bacterial diversity (see **S1 Fig**). The Cliffs Delta statistic for the Observed, ACE, Chao1 and Good's coverage indices confirmed substantial difference in bacterial composition between the two groups (see S1 Table for details).

Beta diversity was assessed using the Bray-Curtis distance matrix and the results were visualized using a PCoA plot. The top two PCoA captured nearly 30% of the variation in diversity (see **Fig 2D**). Then, the PERMANOVA test confirmed significant differences in bacterial composition between BC patients and healthy individuals ($F = 26.944$, $p$-value = 0.001). Additionally, ROC analysis was conducted to evaluate the predictive value of OTUs in distinguishing BC patients from healthy individuals (see **Fig 2E**). These results suggested significant bacterial discrepancies between the BC and healthy groups.

### 3.3 Taxonomic analysis and identification of differentially abundant bacteria

Diversity analyses confirmed significant differences in bacterial abundances between BC patients and healthy controls. To identify dominant bacterial taxa between the groups, a taxonomy analysis was performed at the phylum and genus levels. This analysis identified 29 bacterial phyla and 247 genera across all samples, with the top 20 most abundant phyla and genera displayed in Fig 3A and 3B.

The bacterial community was predominantly composed of Bacteroidetes, Firmicutes, and Proteobacteria (see **Fig 3A**). Firmicutes were less abundant in BC patients than in healthy controls (34.34% vs 40.54%), while Proteobacteria were more abundant (13.14% vs 9.32%). Bacteroidetes had nearly equal abundance in both groups (see **Fig 3C**). Other bacterial phyla showed insignificant changes between the groups. At the genus level, the dominated taxa including *Prevotella*, *Bacteroides*, *Roseburia*, *Succinivibrio*, *Ruminococcus*, and *Blautia* (see **Fig 3B**). *Prevotella*, *Roseburia*, *Succinivibrio*, and *Ruminococcus* were less abundant in BC patients compared to healthy controls, while *Bacteroides* and Blautia were significantly more abundant in BC patients (see **Fig 3D**). However, *Bacteroides*, from the phylum *Bacteroidetes*, emerged as the dominant genus in BC patients.

Using ZIGMM model, 187 DABT were identified between the groups ($p$-value<0.05). Among these, 26 highly differential genera with $\log_2$FC>2.0, $\log_2$FC<-2.0 and $p$-value<0.05, were identified (see **Table 1**). Of the 26 genera, 24 were less abundant in BC patients, meaning they were downregulated, while only two (*Prevotella* and *Anaerovibrio*) were more abundant meaning they were upregulated. At the species level, *Prevotella copri* was upregulated, whereas species such as *Eggerthella lenta*, *Bacteroides uniformis*, *Bacteroides ovatus*, *Clostridium difficile*, *Clostridium saccharogumia*, and *Clostridium citroniae* were downregulated. Additionally, five phyla (Actinobacteria, Bacteroidetes, Firmicutes, Proteobacteria, and Verrucomicrobia) corresponding to these genera were differentially abundant between the groups. These identified bacterial taxa were used for further functional analysis to detect bKGs. These findings suggest that these differential bacteria may play a role in co-infections and potentially contribute to the development of BC in patients.

### 3.4 Pathway-based gene identification from DABT

PICRUSt2 analysis was employed to predict pathway based BC-causing genes from the 26 DABT identified by ZIGMM to understand the functional role of BC-related bacteria. This functional analysis compared the estimated functional gene abundances associated with KEGG (Kyoto Encyclopedia of Genes and Genomes) orthologs. Significant mean percentage

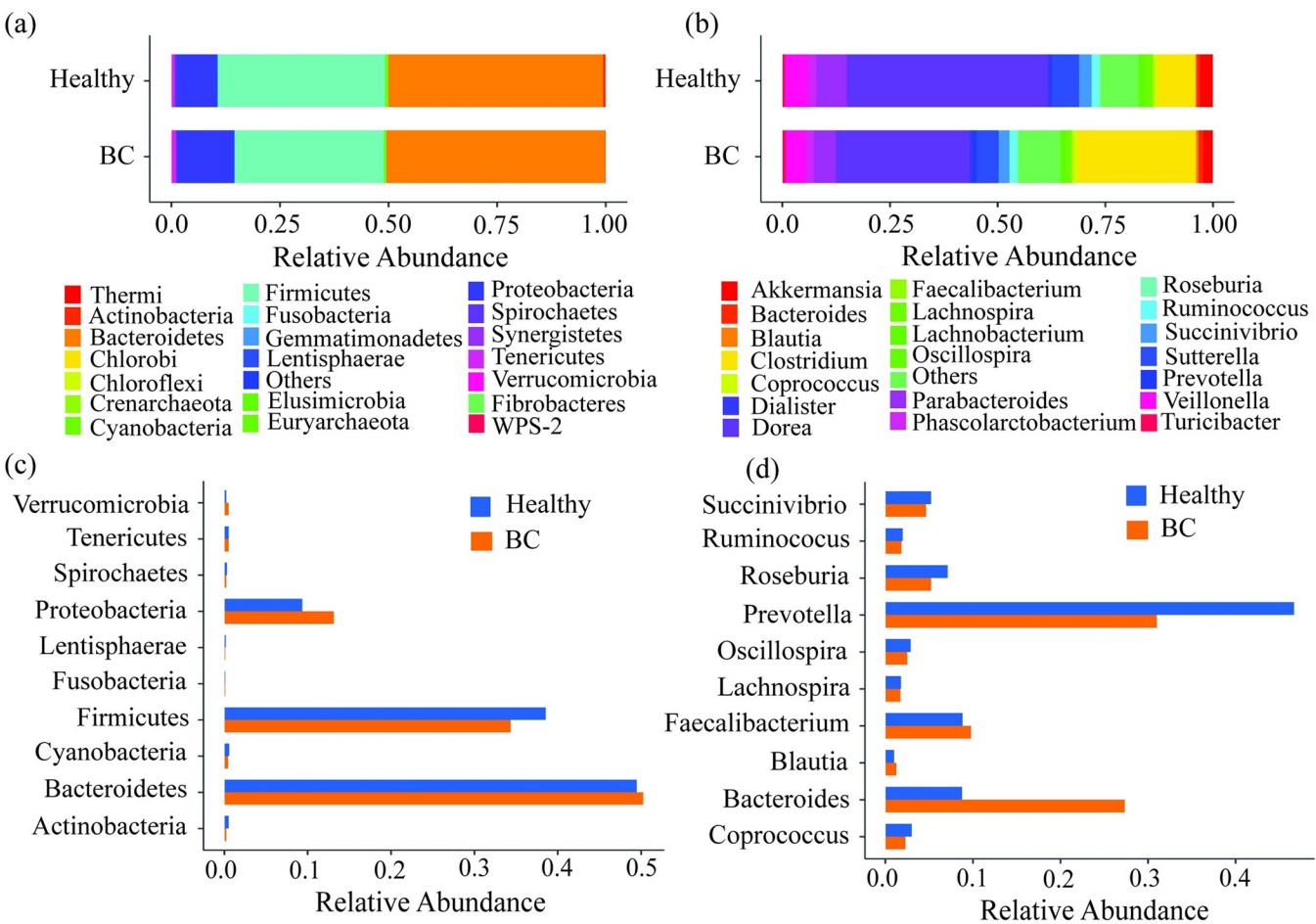

**Fig 3. Bacterial composition at the phylum and genus levels in BC patients and healthy controls.** (a) Bar plot displaying the most abundant bacterial phyla. (b) Bar plot showing the most abundant bacterial genera, with less abundant genera labeled as "others". (c) Bar plot of the top 10 most abundant phyla in both groups. (d) Bar plot of top 10 bacterial genera across the groups.

differences between the BC and healthy groups were observed across 19 MetaCyc signaling pathways (see **Fig 4A**).

Among these, twelve pathways including glycolysis I, pyruvate fermentation to propanoate I, purine nucleotides de novo biosynthesis II, and the TCA cycle VI, significantly increased in BC patients. In contrast, seven pathways, such as gondoate biosynthesis, fatty acid elongation (saturated), stearate biosynthesis II, and palmitoleate biosynthesis I, significantly decreased in the BC group. A random forest classifier was then utilized to evaluate the predictive accuracy based on these differential pathways, achieving approximately 72% accuracy (see Fig 4B). Additionally, 165 genes corresponding to the 12 DABT were identified (see S2 Table). Of these, the top 10 bKGs (*mdh*, *pykF*, *gapA*, *zwf*, *pgi*, *tpiA*, *pgk*, *pfkA*, *ppsA*, and *pykA*) were identified by using protein-protein interaction network analysis. These bKGs were enriched to four pathways (see S3 Table), and used them as drug targets (see Fig 4C).

The Spearman correlation analysis between metabolic pathways and DABT reveals significant associations, where specific genera such as *Prevotella copri* and *Bacteroides uniformis* showed strong positive correlations with pathways like "thiamine salvage II" and "S-adenosyl-L-methionine cycle I" indicating their potential role in enhancing these metabolic activities.

**Table 1. Top 26 highly differentially abundant genera between the BC and healthy group ($p$-value$<$0.05).** A log$_2$FC$<$-2 value indicates the genera were downregulated while a Log$_2$FC$>$2 indicates unregulated.

| Phylum | Family | Genus | Species | Log$_2$FC value | Adj. p-value |
|---|---|---|---|---|---|
| Actinobacteria | Coriobacteriaceae | *Eggerthella* | *lenta* | -3.82 | 0.00 |
| Bacteroidetes | Bacteroidaceae | *Bacteroides* | *uniformis* | -3.69 | 0.00 |
| | | *Bacteroides* | *fragilis* | -2.91 | 0.00 |
| | | *Bacteroides* | *ovatus* | -3.31 | 0.00 |
| | | *Bacteroides* | *acidifaciens* | -2.65 | 0.01 |
| | | *Bacteroides* | *plebeius* | -2.75 | 0.01 |
| | Porphyromonadaceae | *Parabacteroides* | *unclassified* | -3.01 | 0.00 |
| | | *Parabacteroides* | *distasonis* | -2.07 | 0.01 |
| | | *Parabacteroides* | *gordonii* | -2.15 | 0.01 |
| | Prevotellaceae | *Prevotella* | *copri* | 2.52 | 0.00 |
| Firmicutes | Peptostreptococcaceae | *Clostridium* | *difficile* | -3.17 | 0.01 |
| | Erysipelotrichaceae | *Eubacterium* | *dolichum* | -2.97 | 0.00 |
| | | *Clostridium* | *saccharogumia* | -3.01 | 0.00 |
| | Lachnospiraceae | *Ruminococcus* | *gnavus* | -2.57 | 0.01 |
| | | *Clostridium* | *citroniae* | -3.79 | 0.01 |
| | | *Clostridium* | *hathewayi* | -2.07 | 0.00 |
| | | *Dorea* | *unclassified* | -2.31 | 0.01 |
| | | *Blautia* | *producta* | -2.99 | 0.01 |
| | Ruminococcaceae | *Faecalibacterium* | *prausnitzii* | -2.51 | 0.01 |
| | | *Oscillospira* | *unclassified* | -2.74 | 0.00 |
| | Veillonellaceae | *Dialister* | *unclassified* | -2.86 | 0.01 |
| | | *Anaerovibrio* | *unclassified* | 2.33 | 0.01 |
| Proteobacteria | Enterobacteriaceae | *Enterobacter* | *unclassified* | -2.32 | 0.00 |
| | | *Shigella* | *sonnei* | -2.76 | 0.00 |
| | Alcaligenaceae | *Sutterella* | *unclassified* | -2.64 | 0.01 |
| Verrucomicrobia | Verrucomicrobiaceae | *Akkermansia* | *muciniphila* | -2.45 | 0.04 |

(see Fig 5). Conversely, DABT like *Eggerthella lenta* exhibit strong negative correlations with the same pathways, suggesting a suppressive effect.

## 3.5 Bacterial key genes (bKGs)-guided drug repurposing

To explore potential candidate drug molecules for BC treatment from a collection of 198 published drugs, molecular docking analysis was performed using our identified bKGs-mediated proteins (*pykA*, *mdh*, *pgi*, *pykF*, *zwf*, *gapA*, *pfkA*, *pgk*, *tpiA*, and *ppsA*) as receptors. The structure of these proteins were downloaded from the PDB with the corresponding IDs *6k0k*, *1ie3*, *3nbu*, *1pky*, *1dpg*, *6utm*, *6pfk*, *1zmr*, *1tmh and 2ols*. These protein structures underwent energy minimization before docking analysis with the processed ligands using AutoDock Vina. The docking analysis provided binding affinity scores (kcal/mol) for each receptor-ligand pair, which were organized in a descending order matrix $A = (A_{ij})$ with receptors as rows and ligands as columns. This binding affinity matrix was visualized by matrix plot (see Fig 6A). Based on these scores, the top 10 drug agents (Digitoxin, Digoxin, Ledipasvir, Suramin, Ergotamine, Venetoclax, Nilotinib, Conivaptan, Dihydroergotamine, and Elbasvir) were selected as candidate drug molecules against our proposed receptors. To validate the proposed drug-target binding performance, molecular docking was performed among the 10 hub-proteins (*BUB1*, *TOP2A*, *CDK1*, *AURKA*, *CDC20*, *EGFR*, *CCNB1*, *CCNA2*, *BUB1B*, and *FN1*) and

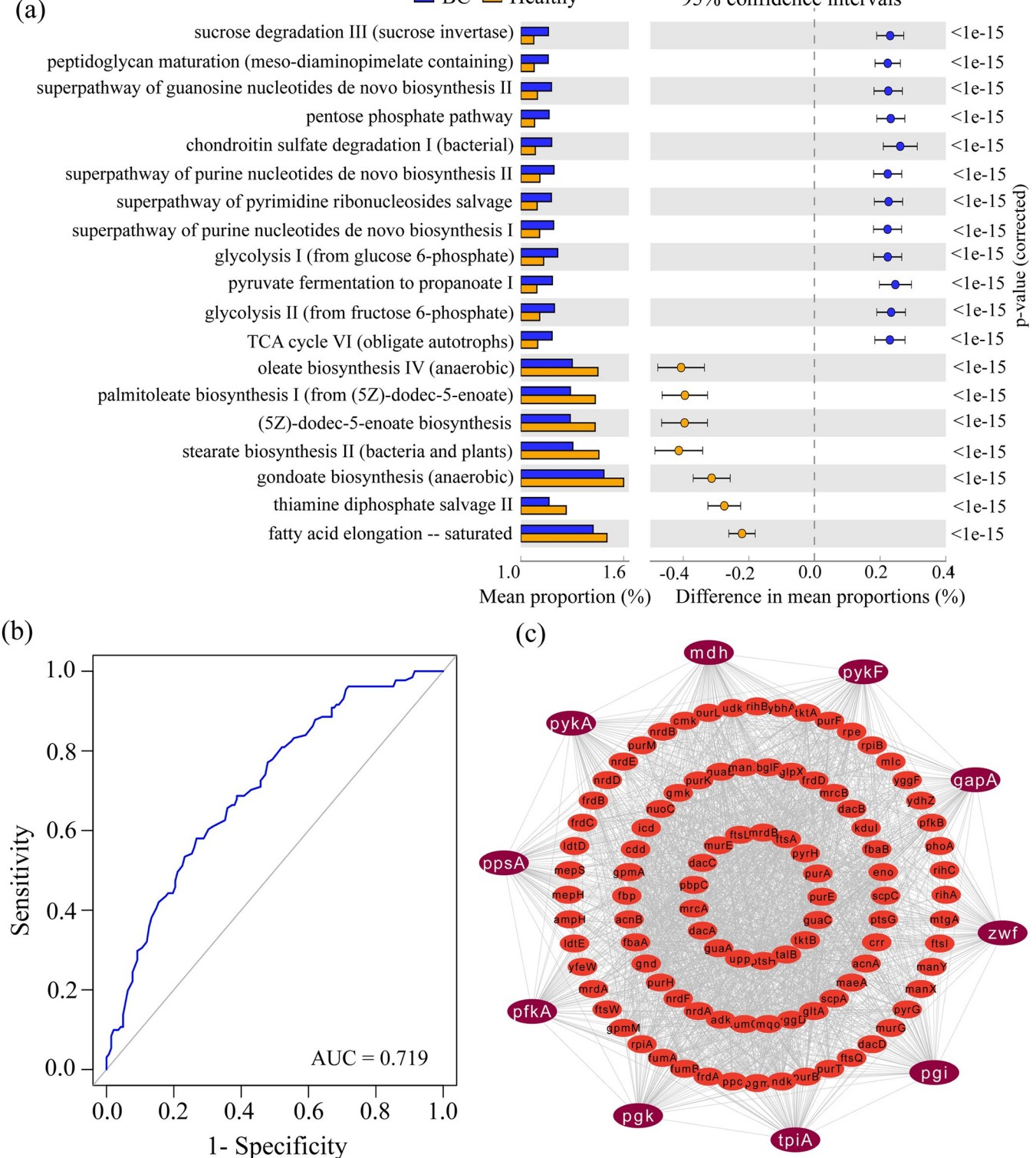

**Fig 4. Exploring key metabolic pathways and bKGs related to BC infection.** (a) Differentially abundant pathways between the BC and healthy individuals. Blue and orange bar indicates the mean proportions of BC and healthy individuals respectively. (b) ROC analysis for the predictive value of BC based on differentially abundant pathways between BC and healthy individuals. (c) Gene interaction analysis of BC related pathways genes to explore the bKGs, while out sided white colour words indicate bKGs.

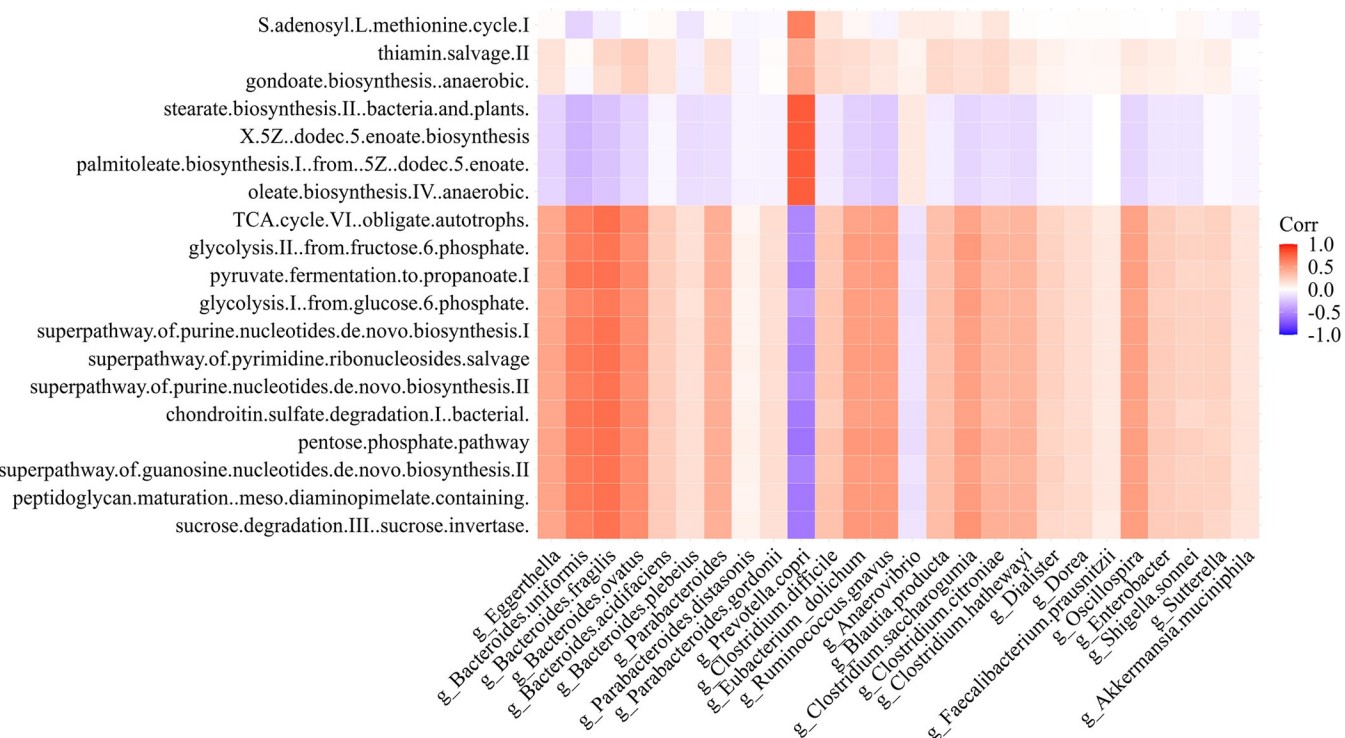

**Fig 5. Spearman correlation analysis was performed between 26 DABT and 19 metabolic pathways, with the x-axis representing the genera and the y-axis representing the metabolic pathways.** The strength of correlations was visualized using color coding: red indicates positive correlations, blue indicates negative correlations, and white indicates no correlation.

collected 198 drug agents. The binding affinity scores were similarly arranged in matrix form and visualized in matrix plot (see Fig 6B). Notably, eight of the top 10 ligands based on binding affinity overlapped with our initial set of 10 lead compounds (see Fig 6A and 6B), suggesting potential effectiveness against BC infections. Therefore, the proposed 10 drug molecules show promise as effective treatments for BC, based on insights from both bKGs mediated receptors and independently identified hub-proteins involved in BC pathogenesis.

Molecular dynamic simulation was performed to evaluate the structural stability of the proposed target-ligand complexes. However, due to limited resources, only first three complexes (Digitoxin-*pykA*, Digoxin-*mdh*, and Ledipasvir-*pgi*) were analysed for structural stability. The root mean square deviations (RMSD) of C-alpha atoms in the docked complexes are presented in Fig 7A. The results demonstrate that the Digoxin-*mdh* complex maintained stability from 10 ns to 100 ns. In contrast, the Digitoxin-*pykA* complex exhibited an initial increase in RMSD up to 40 ns, followed by a decrease at 48 ns, and achieved stability from 60 ns to 80 ns. The Ledipasvir-*pgi* complex showed higher flexibility between 5 ns to 30 ns but reached stability after 55 ns maintaining it until 100 ns. Overall, all three complexes reached a steady state after 55 ns for the remainder of the simulation period. RMSD values for the Digitoxin-*pykA* and Digoxin-*mdh* complexes were consistently below 3.5 Å, indicating stable and rigid structures.

Further stability assessment were conducted using the radius of gyration (Rg) and solvent-accessible surface area (SASA), as depicted in Fig 7C and 7D. The Ledipasvir-*pgi* complex exhibited a higher Rg, suggesting greater flexibility, whereas the Digoxin-*mdh* complex showed a lower Rg, indicating enhanced stability. SASA values for all three complexes were low, indicating a compact surface area. Furthermore, binding energies of the complexes were calculated

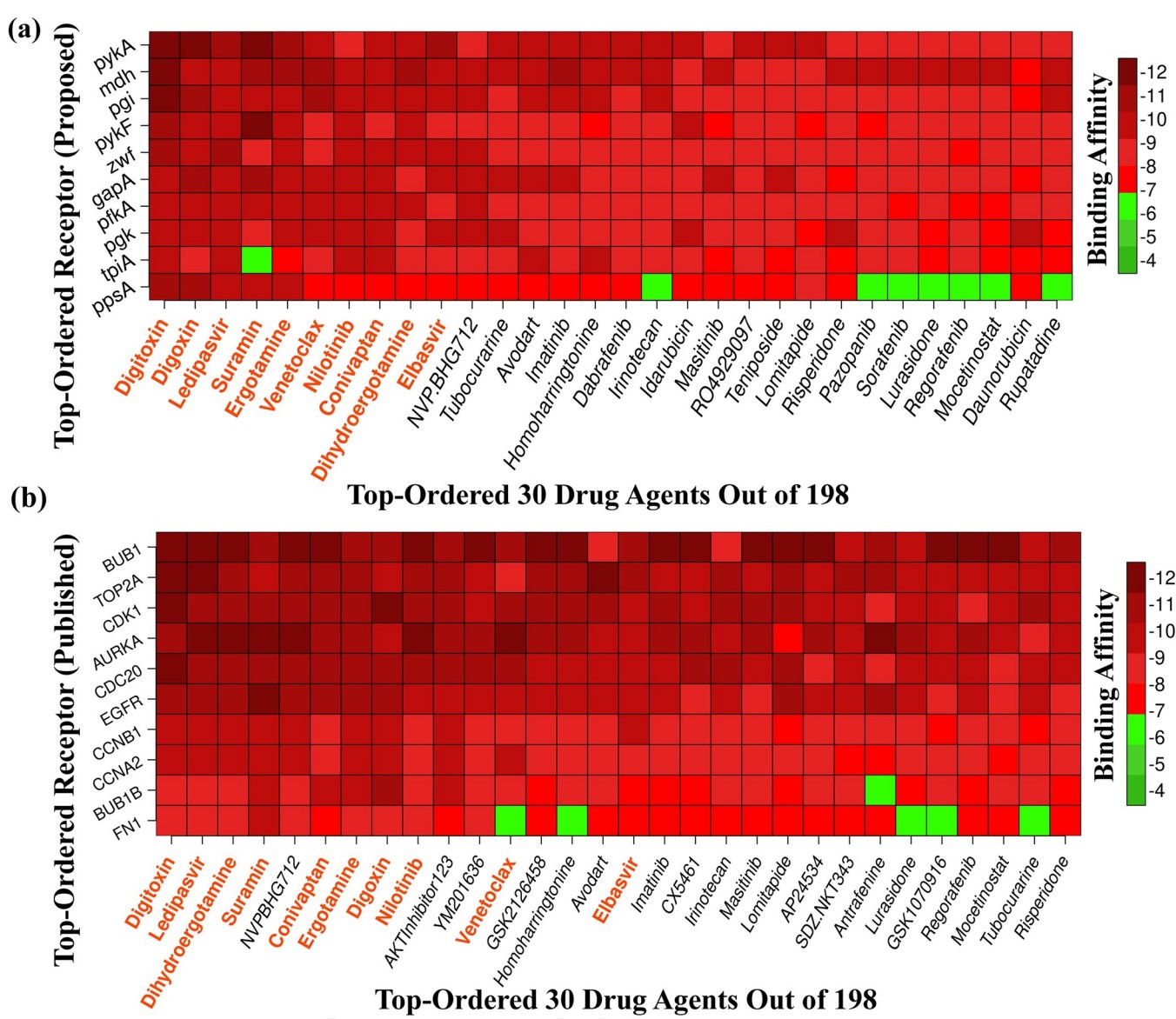

**Fig 6. The matrices displaying protein-ligand binding affinity scores are structured with the X-axis representing the top 30 ranked drug molecules and the Y-axis indicating the ordered bacterial kinase-mediated proteins (bKGs).** Figure (a) illustrates the binding affinities between the proposed 10 bKGs and the top 30 ranked drug molecules, while Figure (b) depicts the binding affinities between the top 10 published key receptors associated with BC and the same set of top 30 ranked drug molecules.

using the MM-PBSA method, with more positive values indicating stronger binding (see Fig 7B). The average binding energies for the Digitoxin-*pykA*, Digoxin-*mdh*, and Ledipasvir-*pgi* complexes were -45.433 kJ/mol, 29.950 kJ/mol, and 135.187 kJ/mol, respectively. Notably, the Ledipasvir-*pgi* complex exhibited the highest binding energy, suggesting stronger ligand binding.

## 4. Discussion

Human microbiota plays a crucial role in the development and progression of various diseases, with notable alterations in composition and functionality observed in conditions such as

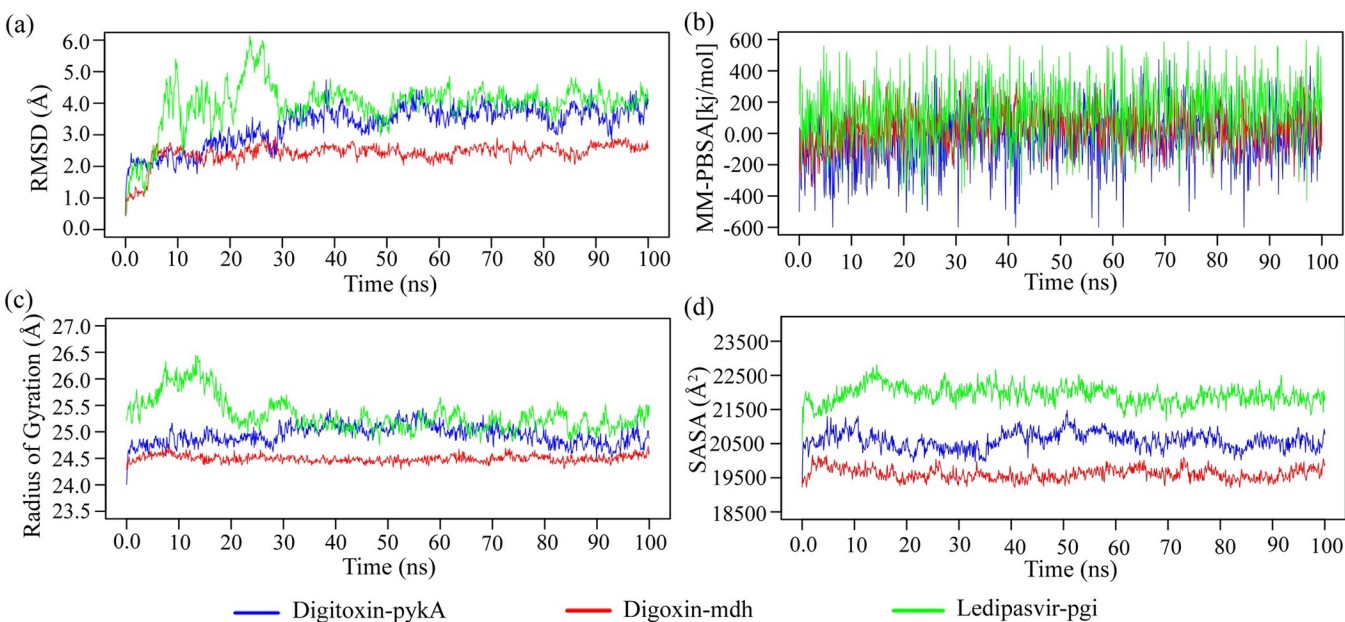

**Fig 7. MD simulation of the top-ranked three docked complexes.** (a) root mean square deviation of the alpha carbon atoms, (b) binding free energy of the complexes where a more positive value indicates better binding, (c) degree of rigidity and compactness analysis of the complexes, (d) protein volume with expansion analysis.

colorectal, liver, prostate, and breast cancers [37, 88]. Epidemiological data suggest that the human microbiota may contribute to approximately 15% of cancer cases globally [89]. Numerous studies have found the link of BC with microbial dysbiosis [17, 88, 90], where specific microorganisms correlate with immune profiles and prognostic indicators [91]. The fecal microbiome has been identified as a potential factor in BC pathobiology [17], highlighting the need for region-specific microbiome profiling in BC patients. This study explored BC-causing bacterial key genes (bKGs) and their inhibitory drug molecules for the Ghanaian female population using in silico analyses to elucidate their potential implications. Our findings reveal a distinct profile of bacterial genetic elements associated with BC, highlighting the diversity and potential pathogenicity of these microbial communities. This study revealed significant differences in bacterial diversity between BC patients and healthy controls, enhancing our understanding of disease pathogenesis and potential diagnostic strategies. Alpha diversity metrics were significantly decrease in in BC patients (p-value < 0.01) [92], suggesting a less diverse microbial community associated with cancer-associated dysbiosis [17]. Good's coverage estimates confirmed the robustness of these findings ($p$-value = 1.5e-10). Beta diversity analysis using the Bray-Curtis distance matrix and PCoA visualization indicated distinct clustering between the groups [92], accounting for nearly 30% of the microbial variation with PERMANOVA test confirming significant differences in bacterial composition ($F$ = 26.944, $p$-value = 0.001). These results align with other studies showing significant differences in the fecal microbial community between BC patients and healthy controls [39, 92, 93].

At the phylum level Bacteroidetes, Firmicutes, and Proteobacteria were predominant, with notable variations in their relative abundances. Firmicutes were less prevalent in BC patients, while *Proteobacteria* were more abundant, indicating a dysbiosis associated with BC, and these results consistent with other studies [18, 94]. The increased abundance of Proteobacteria in both breast tumors and adjacent histologically normal tissues has been reported [95]. The genus-level analysis revealed dominance of *Prevotella*, *Bacteroides*, *Roseburia*, *Succinivibrio*,

*Ruminococcus*, and *Blautia*. Prevotella and other genera showed reduced abundance in BC patients, whereas *Bacteroides* and *Blautia* were significantly enriched. Notably, *P.copri* was found to be enriched in the gut microbiota of BC patients, consistent with our results [96]. These findings suggest a potential role of these genera in infections or direct involvement in BC pathogenesis. *P.corpi* and *B.uniformis* showed strong positive correlations with several pathways, indicating their potential role in modulating metabolic activities conducive to BC progression. Enterotoxigenic *B.fragilis* (ETBF) has been linked to metastasis in BC [97], while *B.ovatus* is associated with favorable treatment outcomes [98]. Although *E.lenta* is usually harmless, it can cause infections in humans and is frequently found in cases of solid organ cancer, diabetes mellitus, and appendicitis [99]. Our study found a significant decrease in *B.uniformis* in BC patients. While not direct link to BC, it is related to obesity and various metabolic functions [100]. *C.difficile*, an anaerobic, gram-positive bacterium primarily colonizing the human colon, poses an elevated risk for cancer patients, resulting in worse outcomes including higher mortality rates [101]. *Eubacterium dolichum*, though not directly implicated in BC, is associated with alterations in the gut microbiome of BC patients [102]. *Blautia producta* has been linked to chronic stress and BC progression, with its reduction potentially contributing to cancer development [103]. *Faecalibacterium prausnitzii*, related to blood metabolites and BC, was reduced in BC patients and negatively correlated with various phosphorylcholines [104].

The differentially abundant bacterial compositions were enriched with 26 metabolic pathways, revealing potential metabolic dysregulations in BC pathophysiology. Our findings highlighted significant alterations in metabolic pathways associated with BC. Twelve pathways, including glycolysis I, pyruvate fermentation to propanoate I, purine nucleotides de novo biosynthesis II, and the TCA cycle VI, were significantly elevated in BC patients, indicating increased metabolic activity crucial for energy metabolism, nucleotide synthesis, and cellular respiration in BC cells [105]. Conversely, seven lipid metabolism pathways, such as gondoate biosynthesis, fatty acid elongation (saturated), stearate biosynthesis II, and palmitoleate biosynthesis I, were downregulated, suggesting alterations in lipid synthesis and fatty acid metabolism [106, 107]. Additionally, 165 KEGG genes corresponding to the 19 differentially abundant signaling pathways were identified. Among these, the top 10 bKGs (*mdh*, *pykF*, *gapA*, *zwf*, *pgi*, *tpiA*, *pgk*, *pfkA*, *ppsA*, and *pykA*) were selected as drug targets by using protein-protein interaction (PPI) network analysis (see **S3 Table**). While the direct impact of PykA and PykF on BC remains uncertain, they are involved in cellular metabolism [108]. The enzyme "*mdh*" is crucial in cancer metabolism, providing an additional source of NAD+ alongside lactate dehydrogenase (LDH) Phosphoglucose isomerase/autocrine motility factor (*PGI/AMF*) is involved in glycolysis and gluconeogenesis and is significantly correlated with BC progression and poor prognosis [109]. *Phosphoglycerate kinase 1* (PGK1), a key enzyme in aerobic glycolysis, is inhibited by miR-16–1-3p in BC patients, repressing cell proliferation, migration, invasion, and metastasis by inhibiting the *PGK1*-mediated Warburg effect [110].

Then top-ranked 10 repurposable drug-molecules (Digitoxin, Digoxin, Ledipasvir, Suramin, Ergotamine, Venetoclax, Nilotinib, Conivaptan, Dihydroergotamine, and Elbasvir) were recommended for inhabiting the BC-causing bKGs through molecular docking analysis, since repurposing approach reduces time and cost compare to the de-novo approach [111–114]. Digitoxin, a cardiac glycoside, shows promise as an anticancer agent [115], while Digoxin has potential effects on BC by inducing ERα degradation, inhibits 17β-estradiol signaling, blocks the cell cycle in the G2 phase, and triggers apoptosis [116]. Ledipasvir, part of the antiviral drug Harvoni, inhibits the BC resistance protein [117]. Venetoclax, a BCL-2 inhibitor, is effective in treating estrogen receptor (ER)-positive and BCL-2-positive metastatic BC [118]. Nilotinib approved for treating chronic myelogenous leukemia (CML) and has shown promise in

BC treatment. Conivaptan was approved for treating hyponatremia, and its antibody-drug conjugates (ADCs) have shown promise in BC treatment by combining monoclonal antibodies with potent cytotoxic agents [119]. Elbasvir, combined with grazoprevir, is FDA-approved for treating chronic hepatitis C [120]. This study provides crucial insights into the unique bacterial and genetic factors influencing BC in the Ghanaian population. So far, this is the first article that explores BC-causing bacterial genes and their inhibitory drug molecules for Ghanaian female population through bioinformatics analysis. However, the findings of this in-silico study requires experimental validation in wet-lab for taking better treatment plan against BC.

## 5. Conclusions

This study analysed 16S rRNA-Seq profiles through bioinformatics approaches to explore breast cancer (BC) causing bacterial key genes (bKGs) and their inhibitory drug molecules for Ghanaian female population. At first, 26 differentially abundant bacterial genera between BC and control groups were detected in which 24 are downregulated and 2 genera (*Prevotella* and *Anaerovibria*) are upregualted in BC patients. Functional enrichment analysis of differential genera identified 19 MetaCyc signaling pathways in which 12 are significantly enriched in BC group by containing 165 genes.Then top-ranked 10 bKGs (*mdh*, *pykF*, *gapA*, *zwf*, *pgi*, *tpiA*, *pgk*, *pfkA*, *ppsA*, and *pykA*) were selected as the targets of candidate drugs through protein-protein interaction (PPI) network analysis. After that, top-ranked 10 drug molecules (Digitoxin, Digoxin, Ledipasvir, Suramin, Ergotamine, Venetoclax, Nilotinib, Conivaptan, Dihydroergotamine, and Elbasvir) were detected as the inhibitors of bKGs by molecular docking analysis. Finally, the binding stability of the top 3 ligands Digitoxin, Digoxin, and Ledipasvir with the targets *mdh*, *pykA* and *pgi* respectively, was investigated through MD simulations and found stable interactions. Thus, the findings of this study could be novel resources for taking an alternative treatment plan against BC after experimental validation in wet-lab.

## Supporting information

**S1 Fig. Good's coverage estimates between BC and healthy control.**
(DOCX)

**S1 Table. Comparison of different microbial communities alpha diversity indices with a significant difference between breast BC (n = 520) and healthy patients (n = 442) based on Wilcoxon-Mann-Whitney test and Cliff's Delta.**
(DOCX)

**S2 Table. Summary of bKGs corresponding to our identified metabolic pathways which are significantly more abundant in BC patients group.**
(DOCX)

**S3 Table. Distribution of identified top 10 bKGs corresponding to their metabolic pathways.**
(DOCX)

**S4 Table. Metadata of host-protein of breast cancer patients obtained by reviewing published articles were used in this study.**
(DOCX)

**S5 Table. Metadata of ligand molecules of breast cancer patients obtained by reviewing published articles were used in this study.**
(DOCX)

## Acknowledgments

Authors would like to acknowledge both reviewers for their valuable comments that were helpful to improve the quality of the manuscript.

## Author Contributions

**Conceptualization:** Md. Kaderi Kibria, Md. Nurul Haque Mollah.

**Data curation:** Md. Kaderi Kibria.

**Formal analysis:** Md. Kaderi Kibria.

**Funding acquisition:** Md. Kaderi Kibria, Md. Nurul Haque Mollah.

**Investigation:** Md. Nurul Haque Mollah.

**Methodology:** Md. Kaderi Kibria.

**Project administration:** Md. Nurul Haque Mollah.

**Resources:** Md. Kaderi Kibria.

**Software:** Md. Kaderi Kibria, Md. Ahad Ali.

**Supervision:** Md. Nurul Haque Mollah.

**Visualization:** Md. Kaderi Kibria.

**Writing – original draft:** Md. Kaderi Kibria.

**Writing – review & editing:** Md. Kaderi Kibria, Md. Ahad Ali, Md. Nurul Haque Mollah.

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
