## [Decision Letter · Decision Letter 0]

13 Aug 2024

PONE-D-24-29239Exploring Bacterial Key Genes and Therapeutic Targets in Breast Cancer among the Ghanaian Population: Insights from In Silico AnalysesPLOS ONE

Dear Dr. Kibria,

Thank you for submitting your manuscript to PLOS ONE. After careful consideration, we feel that it has merit but does not fully meet PLOS ONE’s publication criteria as it currently stands. Therefore, we invite you to submit a revised version of the manuscript that addresses the points raised during the review process.

We look forward to receiving your revised manuscript.

Kind regards,

Palash Mandal

Academic Editor

PLOS ONE

Journal Requirements:

2. Please note that PLOS ONE has specific guidelines on code sharing for submissions in which author-generated code underpins the findings in the manuscript. In these cases, we expect all author-generated code to be made available without restrictions upon publication of the work. 

Please review our guidelines at https://journals.plos.org/plosone/s/materials-and-software-sharing#loc-sharing-code and ensure that your code is shared in a way that follows best practice and facilitates reproducibility and reuse.

**Additional Editor Comments:**

Dear Authors,

Thank you for submitting your manuscript to PLOS ONE. After careful consideration, we feel that it has merit but does not fully meet PLOS ONE publication criteria as it currently stands. The shortcomings of this paper needs to be worked out before it can be considered for publication. Therefore, we invite you to resubmit a revised version of the manuscript that addresses the points raised during the review process.

For your guidance, the reviewers' comments are included below.

Thank you for giving us the opportunity to consider your work. Specific concerns expressed during peer review were:

*Comments from Reviewer 1*

In this study authors identified 26 differential genera between BC and healthy groups of bacterial sequences. They suggested top 10 bKGs and bKG-guided top-ranked 10 drug molecules that could be potential treatment strategies for bacterial infections in BC patients. The manuscript seems interesting; however, some minor revision is required before it can be published in PLOS ONE.

1, Authors can add statistics of breast cancer in Africa in the introduction section.

2, Workfellows should be mentioned at the end of the introduction section.

3, Throughout the paper, terms such as Breast Cancer (BC) were repeatedly written in full form; instead, use the full form only once and consistently use the abbreviation (e.g., BC) in subsequent mentions.

4, Similarly, some other terms such as identifying bacterial key genes (bKGs) were repeatedly written in full form as well as and also check others terms.

5, We collected 16S rRNA sequence data from the NCBI while 520 samples were BC patients and 442 were healthy controls.

Avoid using “we” when writing paper or thesis; just make sentences in passive voice.

6, we explored top 10 bKGs (mdh, pykF, gapA, zwf, pgi, tpiA, pgk, pfkA, ppsA, and pykA) through protein-protein interactions analysis as potential pathogenic factors.

Please replace the use of “we” in the sentence as well as the previous one.

ATTENTION: There are too many “we” that been used in this paper from abstract to results and discussion, thus, please recheck and replace it with better option so that the manuscript would sounds finer.

7, in Figure 5, you should mention what you represent in Y-axis and X-axis like in figure 6a.

8, Authors protein structures were obtained from the Protein Data Bank (PDB) database, PDB IDs of the proteins used for the docking needs to be mentioned.

9, Author suggests to make three-line format tables.

10, Go through the entire manuscript, looking for any typos and grammatical issues. Clarity of statements should be checked in several places.

11, Author identified 26 differential genera between BC and healthy groups. How they specifically mentioned later that the two genera Prevotella and Anaerovibria were significantly enriched among them in BC patients? In lines 32-33

*Comments from Reviewer 2*

The manuscript, titled “Exploring Bacterial Key Genes and Therapeutic Targets in Breast Cancer among the Ghanaian Population: Insights from In-Silico Analyses,” describes breast cancer (BC) as a global health issue, with microbiota and epigenetic changes being significant risk factors. A study in Ghana identified bacterial key genes (bKGs) associated with BC infections and explored potential drug molecules targeting these bKGs. The study found significant bacterial discrepancies between BC and healthy groups, with Prevotella and Anaerovibria genera enriched in BC patients. The top 10 bKGs were identified as potential pathogenic factors, and drug molecules were identified using molecular docking analysis. Following are some suggestions that, in general, improve the presentation and quality of this work:

1. The authors should revise the title of this manuscript. Please make sure that selecting a research manuscript topic with qualities like originality, relevance, specificity, feasibility, interest, impact potential, interdisciplinary relevance, ethical considerations, clarity, and contribution to knowledge significantly impacts its success.

2. The abstract contains excessive detail and should be rewritten to clarify the rationale for mentioning the study’s objectives in a more prominent manner and remove redundant expressions. For instance, see lines 29–31: “To do this, we collected 16S rRNA sequence data... controls.”. Please justify how you collect the sequence data. The word “”collected” seems to mismatch with this information. Please re-check the use of some words like line 31 ('discrepancies'), line 36 ('explored'), line 43 (‘valuable insight'), ‘potential treatment strategies', etc.

3. Keywords: A few of them should not be included at this level. The authors should re-check the suitability of this section and only add the prominent keywords that are significant to this work, which will help the global readers appropriately, select this work in the future.

4. Introduction: The introduction isn’t organized and looks more like a mishmash of information, with several grammar and punctuation errors. Authors should comprehensively revise this section, as a good manuscript introduction should provide clear context, highlight the study's importance, address the specific research problem, summarize the current state of research, justify the research approach, define the study's scope and focus, be engaging and accessible, and outline the structure of the manuscript. For instance, “The World Health Organization reported 2.3 million women were diagnosed with BC in 2020, resulting in 68,500 deaths worldwide, with new cases expected to rise to 3.06 million annually by 2040” needs a proper citation. Lines 55–56, the mentioned statement has some confusion; kindly re-check and revise accordingly. Please re-check lines 65–67. Similarly, please justify why you mentioned lines 80–83. Introduction should also provide a literature review, justify the research approach, and provide a clear narrative that aligns with the study's objectives. Unfortunately, the present state of the manuscript does not address the above attributes of the introduction. What do you mean by the statement ‘valuable insight’? Avoid jargon and overly technical language for a broad audience (please revise lines 95–96). Incorporating these elements will help create a well-rounded and effective introduction that captures the reader’s interest and clearly communicates the significance and direction of the research.

5. Methodology: sub-headings 2.1 and 2.2 need to be revised with more detail and explanation. Similarly, the content described in the 2.1 sections needs more description with thorough literature. I will suggest here to further break this sub-heading accordance to work relevance (optional). What was the justification for using or selecting published host-protein and ligand molecules of BC patients as metadata for this work? Lines 114–116: “Additionally, a total of 198 drug compounds associated with BC were collected by reviewing 17 articles.” Please cite some authentic references in support of this activity. Details mentioned in lines 120–129 need to be more organized. In 2.2.2., please justify the use of Principle Coordinate Analysis (PCoA) plots. In 2.2.3., why do the mentioned statistical analyses include the Wilcoxon signed rank test, the Mann-Whitney U test, and the Kruskal-Wallis test (please justify)? Lines 155–157 need to be more organized according to work done. Please also explain in detail: “Genes corresponding to these pathways were compiled.” Sub-heading 2.2.5. also needs more clarity in mentioning docking and simulation aspects. What is the reason for mentioning sub-heading 2.2.6.? Is this really a requirement for bioinformatics-based research? Justify.

6. The attached work flow is of poor quality and visual. Authors should avoid this limited and primary-level work flow. I would suggest the authors add an appealing workflow with visuals, as shown in a variety of manuscripts, to enhance the understanding of their work. Please check the following manuscripts for a better reference.

DOI: 10.1021/acsomega.2c04871

DOI: 10.1021/acsptsci.2c00212

DOI: 10.1021/acs.jcim.0c00488

7. Results: The output seems satisfactory; however, the language used by the authors when describing their outcomes frequently overstates the conclusions that can be drawn based solely on computational studies lacking experimental data. Most sub-headings in these sections, along with the subsequent details, tend to overstate facts. Currently, this part of the manuscript is wordy and contains some redundancies. For instance, sub-heading 3.3 needs more improvement with relevant details. What is the purpose of beta-diversity analysis (justify)? In Table 1, please mention the inference you extracted from the LogFC value analysis (for the readers) regarding genes selection. With reference to Figure 5, a Pearson correlation analysis was conducted between the 26 differentially abundant genera and 19 metabolic pathways. Based on this fact, Pearson correlation is more sensitive to outliers, which can significantly affect the correlation coefficient since it relies on the mean and standard deviation. Whereas Spearman correlation is less sensitive to outliers as it uses ranks instead of actual data values, reducing the impact of extreme values. The author should justify the use of Pearson correlation analysis. Data are presented in figures and left to the reader to deduce their relevance and appropriateness. The details available in lines 275-278 and 280-282 need more clarity. Figures 5 and 6 need more discussion on the basis of their outcomes. Similarly, in Figure 7, the comparative discussion needs to be incorporated from the MD simulation analysis. While the manuscript is good, it lacks breadth, largely due to its verbosity and vagueness in the Results and Discussion sections. I believe a rewrite would strengthen this manuscript.

8. Conclusion: In its present form, it is very simple and monotonous with the work scope. It will be better if this section is carefully revised.

9. Strengths and limitations: It will be better if you specifically write them in your manuscript.

10. References: If possible, the author should provide some of the of the latest references in the introduction and discussion sections.

11. Other: Some English grammar mistakes exist; please further check the manuscript and refine the language carefully.

Reviewers' comments:

Reviewer's Responses to Questions

**Comments to the Author**

1. Is the manuscript technically sound, and do the data support the conclusions?

Reviewer #1: Yes

Reviewer #2: Yes

2. Has the statistical analysis been performed appropriately and rigorously? 

Reviewer #1: Yes

Reviewer #2: Yes

3. Have the authors made all data underlying the findings in their manuscript fully available?

Reviewer #1: No

Reviewer #2: Yes

4. Is the manuscript presented in an intelligible fashion and written in standard English?

Reviewer #1: Yes

Reviewer #2: Yes

5. Review Comments to the Author

Reviewer #1: In this study authors identified 26 differential genera between BC and healthy groups of bacterial sequences. They suggested top 10 bKGs and bKG-guided top-ranked 10 drug molecules that could be potential treatment strategies for bacterial infections in BC patients. The manuscript seems interesting; however, some minor revision is required before it can be published in PLOS ONE.

1, Authors can add statistics of breast cancer in Africa in the introduction section.

2, Workfellows should be mentioned at the end of the introduction section.

3, Throughout the paper, terms such as Breast Cancer (BC) were repeatedly written in full form; instead, use the full form only once and consistently use the abbreviation (e.g., BC) in subsequent mentions.

4, Similarly, some other terms such as identifying bacterial key genes (bKGs) were repeatedly written in full form as well as and also check others terms.

5, We collected 16S rRNA sequence data from the NCBI while 520 samples were BC patients and 442 were healthy controls.

Avoid using “we” when writing paper or thesis; just make sentences in passive voice.

6, we explored top 10 bKGs (mdh, pykF, gapA, zwf, pgi, tpiA, pgk, pfkA, ppsA, and pykA) through protein-protein interactions analysis as potential pathogenic factors.

Please replace the use of “we” in the sentence as well as the previous one.

ATTENTION: There are too many “we” that been used in this paper from abstract to results and discussion, thus, please recheck and replace it with better option so that the manuscript would sounds finer.

7, in Figure 5, you should mention what you represent in Y-axis and X-axis like in figure 6a.

8, Authors protein structures were obtained from the Protein Data Bank (PDB) database, PDB IDs of the proteins used for the docking needs to be mentioned.

9, Author suggests to make three-line format tables.

10, Go through the entire manuscript, looking for any typos and grammatical issues. Clarity of statements should be checked in several places.

11, Author identified 26 differential genera between BC and healthy groups. How they specifically mentioned later that the two genera Prevotella and Anaerovibria were significantly enriched among them in BC patients? In lines 32-33

Reviewer #2: The manuscript, titled “Exploring Bacterial Key Genes and Therapeutic Targets in Breast Cancer among the Ghanaian Population: Insights from In-Silico Analyses,” describes breast cancer (BC) as a global health issue, with microbiota and epigenetic changes being significant risk factors. A study in Ghana identified bacterial key genes (bKGs) associated with BC infections and explored potential drug molecules targeting these bKGs. The study found significant bacterial discrepancies between BC and healthy groups, with Prevotella and Anaerovibria genera enriched in BC patients. The top 10 bKGs were identified as potential pathogenic factors, and drug molecules were identified using molecular docking analysis. Following are some suggestions that, in general, improve the presentation and quality of this work:

1. The authors should revise the title of this manuscript. Please make sure that selecting a research manuscript topic with qualities like originality, relevance, specificity, feasibility, interest, impact potential, interdisciplinary relevance, ethical considerations, clarity, and contribution to knowledge significantly impacts its success.

2. The abstract contains excessive detail and should be rewritten to clarify the rationale for mentioning the study’s objectives in a more prominent manner and remove redundant expressions. For instance, see lines 29–31: “To do this, we collected 16S rRNA sequence data... controls.”. Please justify how you collect the sequence data. The word “”collected” seems to mismatch with this information. Please re-check the use of some words like line 31 ('discrepancies'), line 36 ('explored'), line 43 (‘valuable insight'), ‘potential treatment strategies', etc.

3. Keywords: A few of them should not be included at this level. The authors should re-check the suitability of this section and only add the prominent keywords that are significant to this work, which will help the global readers appropriately, select this work in the future.

4. Introduction: The introduction isn’t organized and looks more like a mishmash of information, with several grammar and punctuation errors. Authors should comprehensively revise this section, as a good manuscript introduction should provide clear context, highlight the study's importance, address the specific research problem, summarize the current state of research, justify the research approach, define the study's scope and focus, be engaging and accessible, and outline the structure of the manuscript. For instance, “The World Health Organization reported 2.3 million women were diagnosed with BC in 2020, resulting in 68,500 deaths worldwide, with new cases expected to rise to 3.06 million annually by 2040” needs a proper citation. Lines 55–56, the mentioned statement has some confusion; kindly re-check and revise accordingly. Please re-check lines 65–67. Similarly, please justify why you mentioned lines 80–83. Introduction should also provide a literature review, justify the research approach, and provide a clear narrative that aligns with the study's objectives. Unfortunately, the present state of the manuscript does not address the above attributes of the introduction. What do you mean by the statement ‘valuable insight’? Avoid jargon and overly technical language for a broad audience (please revise lines 95–96). Incorporating these elements will help create a well-rounded and effective introduction that captures the reader’s interest and clearly communicates the significance and direction of the research.

5. Methodology: sub-headings 2.1 and 2.2 need to be revised with more detail and explanation. Similarly, the content described in the 2.1 sections needs more description with thorough literature. I will suggest here to further break this sub-heading accordance to work relevance (optional). What was the justification for using or selecting published host-protein and ligand molecules of BC patients as metadata for this work? Lines 114–116: “Additionally, a total of 198 drug compounds associated with BC were collected by reviewing 17 articles.” Please cite some authentic references in support of this activity. Details mentioned in lines 120–129 need to be more organized. In 2.2.2., please justify the use of Principle Coordinate Analysis (PCoA) plots. In 2.2.3., why do the mentioned statistical analyses include the Wilcoxon signed rank test, the Mann-Whitney U test, and the Kruskal-Wallis test (please justify)? Lines 155–157 need to be more organized according to work done. Please also explain in detail: “Genes corresponding to these pathways were compiled.” Sub-heading 2.2.5. also needs more clarity in mentioning docking and simulation aspects. What is the reason for mentioning sub-heading 2.2.6.? Is this really a requirement for bioinformatics-based research? Justify.

6. The attached work flow is of poor quality and visual. Authors should avoid this limited and primary-level work flow. I would suggest the authors add an appealing workflow with visuals, as shown in a variety of manuscripts, to enhance the understanding of their work. Please check the following manuscripts for a better reference.

DOI: 10.1021/acsomega.2c04871

DOI: 10.1021/acsptsci.2c00212

DOI: 10.1021/acs.jcim.0c00488

7. Results: The output seems satisfactory; however, the language used by the authors when describing their outcomes frequently overstates the conclusions that can be drawn based solely on computational studies lacking experimental data. Most sub-headings in these sections, along with the subsequent details, tend to overstate facts. Currently, this part of the manuscript is wordy and contains some redundancies. For instance, sub-heading 3.3 needs more improvement with relevant details. What is the purpose of beta-diversity analysis (justify)? In Table 1, please mention the inference you extracted from the LogFC value analysis (for the readers) regarding genes selection. With reference to Figure 5, a Pearson correlation analysis was conducted between the 26 differentially abundant genera and 19 metabolic pathways. Based on this fact, Pearson correlation is more sensitive to outliers, which can significantly affect the correlation coefficient since it relies on the mean and standard deviation. Whereas Spearman correlation is less sensitive to outliers as it uses ranks instead of actual data values, reducing the impact of extreme values. The author should justify the use of Pearson correlation analysis. Data are presented in figures and left to the reader to deduce their relevance and appropriateness. The details available in lines 275-278 and 280-282 need more clarity. Figures 5 and 6 need more discussion on the basis of their outcomes. Similarly, in Figure 7, the comparative discussion needs to be incorporated from the MD simulation analysis. While the manuscript is good, it lacks breadth, largely due to its verbosity and vagueness in the Results and Discussion sections. I believe a rewrite would strengthen this manuscript.

8. Conclusion: In its present form, it is very simple and monotonous with the work scope. It will be better if this section is carefully revised.

9. Strengths and limitations: It will be better if you specifically write them in your manuscript.

10. References: If possible, the author should provide some of the of the latest references in the introduction and discussion sections.

11. Other: Some English grammar mistakes exist; please further check the manuscript and refine the language carefully.

6. PLOS authors have the option to publish the peer review history of their article (what does this mean?). If published, this will include your full peer review and any attached files.

Reviewer #1: **Yes: **MD. SHAHIN ALAM

Reviewer #2: No

---

## [Author Response · Author response to Decision Letter 0]

16 Sep 2024

Reviewer 1, Comment 1: Authors can add statistics of breast cancer in Africa in the introduction section.

Response: Thank you for your suggestion. We revised the introduction section by including statistical data about breast cancer in Africa.

---

## [Decision Letter · Decision Letter 1]

8 Oct 2024

Exploring Bacterial Key Genes and Therapeutic Agents for Breast Cancer among the Ghanaian Female Population: Insights from In Silico Analyses

PONE-D-24-29239R1

Dear Dr. Md. Kaderi Kibria,

We’re pleased to inform you that your manuscript has been judged scientifically suitable for publication and will be formally accepted for publication once it meets all outstanding technical requirements.

Kind regards,

Palash Mandal

Academic Editor

PLOS ONE

---

## [Editor Report · Acceptance letter]

13 Nov 2024

PONE-D-24-29239R1 

PLOS ONE

Dear Dr. Kibria, 

I'm pleased to inform you that your manuscript has been deemed suitable for publication in PLOS ONE. Congratulations! Your manuscript is now being handed over to our production team.

Kind regards, 

on behalf of

Prof. Palash Mandal 

Academic Editor

PLOS ONE